# FEDERATED BRAIN TUMOUR SEGMENTATION USING MULTI-MODAL INFORMATION FUSION

## ABSTRACT

Federated learning applications to privacy-sensitive domain involving medical data is one of the important directions of research in recent times. We introduce a federated framework for multi-modal brain tumour segmentation that integrates hierarchical cross-modal fusion with end-to-end distributed training under simulated inter-site heterogeneity. Our architecture couples a shared encoder with correlation-aware, attention-based fusion across MRI modalities (T1, T1CE, T2, FLAIR) and multi-scale deep supervision, addressing a key gap in multimodal fusion under federated constraints where modality and site distributions are non-IID and communication budgets are limited. To accelerate convergence and reduce communication, we initialize the federated backbone from a pretrained network, a strategy known to stabilize optimization and decrease rounds to target accuracy in FL. Empirically, the proposed method delivers consistent gains on Enhancing Tumor (ET), Tumor Core (TC) and Mean Dice over strong centralized baselines, while achieving comparable performance on Whole Tumor (WT), indicating that cross-modal reasoning primarily benefits the more challenging, heterogeneous subregions. We observe faster federated convergence attributable to pretrained initialization, supporting the practicality of our approach for resource-constrained clinical deployments. Collectively, these results advance multimodal fusion in federated neuro-oncology and provide a rigorous, large-scale evaluation.

## 1 INTRODUCTION

Segmentation of brain tumours remains a significant challenge in medical imaging, directly impacting the quality of diagnosis and treatment strategies. The growing adoption of machine learning techniques, particularly deep learning models, has revolutionized segmentation tasks; however, these advancements often hinge on access to large, diverse datasets, thereby raising serious privacy and data sovereignty concerns. Federated learning (FL) has recently emerged as a transformative strategy, enabling institutions to jointly train AI models without sharing sensitive data, offering a solution to prevalent privacy barriers. Yet, while FL addresses the data governance issue, segmentation accuracy is often hampered by the heterogeneity in data distributions which causes different clients $\mathcal{C}_i$ in the federated learning setup to lie in different parameter state $\mathcal{P}_i$ and by the limited exploitation of the rich information contained in multiple imaging modalities.

Federated learning (FL) often faces challenges such as *data scarcity* and *non-IID data distributions* across clients. *Communication efficiency* and *limitations of computational resources* are other parameters which should be considered in the federated learning scenario. A larger client model will require a larger bandwidth in each communication round, even if not all clients participate in a particular communication round. Past centralized baselines like Residual 3D UNet Raza et al. (2023), UNETR Hatamizadeh et al. (2022), TransBTS Wang et al. (2021), SwinUNETR Hatamizadeh et al. (2021), nnFormer Zhou et al. (2023) used large datasets and were trained for a large number of epochs. However, to reduce the communication and computational overhead, initializing the global model with a pretrained network provides rich and generic feature representations, enabling faster and more stable convergence. Furthermore, leveraging pretrained models typically leads to improved generalization and accuracy, especially when client data is limited or imbalanced. The pretrained model also results in better communication efficiency as it requires fewer communication rounds to achieve convergence, without compromising client data privacy.

In this paper, we utilize the strengths of federated learning and multi-modal fusion. Our framework (*FedUNetMM*) leverages advanced attention-based fusion techniques to integrate multi-modal signals in a federated context, benchmarking its performance on the benchmark datasets such as BraTS 2018, BraTS 2019, and BraTS 2020 Menze et al. (2015); Bakas et al. (2017; 2018). We systematically evaluate segmentation outcomes, the potential of multi-modal federated AI for clinical neuroimaging, and the effects of different positional embeddings on the segmentation performance.

The major contributions of this work can be summarized as follows:

- We propose a novel 2D convolutional architecture based on multi-modal fusion to incorporate information from multiple magnetic resonance sequences for the segmentation of different brain tumour features.

- We also propose a novel attention-based multi-modal fusion architecture for amalgamating information from the four sequences, T1, T1CE, T2, and FLAIR.

- We show that it is possible to obtain performance better than or at par performance to centralized baseline trained from scratch for hundreds of epochs on large centralized datasets with much less number of parameters.

- We also study the effect of different types of spatial positional embeddings on the brain tumor segmentation performance in the federated learning scenario.

- We conduct a comprehensive evaluation on three benchmark datasets, BraTS2018, BraTS2019, and BraTS2020.

## 1.1 RELATED WORKS

**Federated Brain Tumour Segmentation:** Early work established the promise of federated learning (FL) for brain tumor segmentation. The work Sheller et al. (2020) showed FedAvg McMahan et al. (2017) closely matched centralized training on BraTS 2018, while cyclic weight transfer suffered catastrophic forgetting. Follow-ups examined differential privacy and local momentum restart Li et al. (2019). The FeTS 2021–2022 challenges Pati et al. (2021) systematically assessed client sampling, server synchronization frequency, and aggregation weighting in FedAvg, and spurred new methods: FedCostwAvg Mächler et al. (2021), which weights updates by sample size and loss improvement, and FedPIDAvg Mächler et al. (2022), which adds a momentum-like term on losses. Participants also explored FedAvg-M, FedAdam, robust aggregations (median, trimmed mean), Top-K mean, and other loss-based schemes Rawat et al. (2022); Tuladhar et al. (2022), as well as adaptive local epochs, learning-rate decay, FedNova, and FedAvg-M Isik-Polat et al. (2022). A real-world study Pati et al. (2022) across 71 institutions demonstrated that FedAvg at scale substantially outperforms models trained only on public BraTS data. However, due to geo-restrictions on the Synapse website, we were unable to download the FeTS 2022 dataset.

**Centralized Brain Tumor Segmentation:** There have been several innovations in CNN or Transformer-based U-Net variants for brain tumor segmentation, including 3D U-Net undefinediçek et al. (2016), Residual 3D U-Net Raza et al. (2023), TransBTS Wang et al. (2021), UNETR Hatamizadeh et al. (2022), SwinUNETR Hatamizadeh et al. (2021), UNETR++ Shaker et al. (2024), and nnFormer Zhou et al. (2023). 3D U-Net established the encoder–decoder with skip connections as a standard for volumetric tumor delineation on benchmarks like BraTS. Residual U-Net variants deepen the network and ease optimization, and exhibits the steady evolution of U-Net families toward stronger multiscale representation and stability. Hybrid CNN–Transformer designs such as TransBTS combine convolutional feature extractors with self-attention to capture longer-range dependencies and cross-modality context beyond pure CNNs. Transformer-centric models (e.g., UNETR, SwinUNETR, nnFormer) pair hierarchical attention encoders with UNet-like decoders to better model 3D context and global relations in medical image segmentation. Further refinements like UNETR++ aim to enhance encoder–decoder interactions and multi-scale fusion within this design space, reflecting the broader trend of progressively improved U-Net-derived architectures.

We adopt a lightweight 2D U-Net in a federated setting to minimize parameters and communication overhead while retaining competitive brain-tumor segmentation quality, and focus on improving the architecture with a standard FedAvg aggregation method.

## 1.2 PRELIMINARIES: FEDERATED LEARNING

Suppose there are $K$ different sources of data stored locally in each client. The datasets are independent but non-identically distributed. The objective in this scenario is to learn a shared model by optimizing the Eqn. 1. By letting each data source $k \in K$ have a local dataset $\mathcal{D}_k = \{x_k^i, y_k^i\}_{i=1}^{|\mathcal{D}_k|}$, where $x_k^i$ and $y_k^i$ denotes the image and its corresponding labels or segmentation masks, the federated learning problem can be formulated as finding a set of parameters which minimizes the following objective:

$$\mathcal{L}(\theta, \alpha) = \sum_{k=1}^{K} \alpha_k \mathcal{L}_k(\theta, \mathcal{D}_k) \tag{1}$$

where $\alpha = [\alpha_1, \alpha_2, \ldots, \alpha_K]$ represents the aggregation weights of the clients and $\mathcal{L}_k$ is the loss for the $k$-th client. The aggregation approach used in this work, is the basic framework FedAvg McMahan et al. (2017), which uses the ratio of the number of samples in the $k$-th client to the sum of the number of samples over all the clients as the aggregation weight, that is, $\lambda_k = \frac{|\mathcal{D}_k|}{\sum_{i=1}^{K} |\mathcal{D}_i|}$.

In this work, we are dealing with a cross-silo federated learning scenario. In each communication round $r$, the local model parameters $w_k^r$ are trained on the local dataset for $E$ number of local epochs. After the training on the client ends, the trained model is communicated to the server. The server then aggregates the client model parameters according to the weights $\alpha$ to give the aggregated model $w_g^r$. This aggregated global model $w_g^r$ is then broadcasted back to the clients and used to initialize the client models for the next communication round or as a reference for further training.

## 2 METHODOLOGY

### 2.1 OVERVIEW OF FRAMEWORK

Motivated by the challenge of reliably delineating heterogeneous brain tumour structures across patient cohorts and institutions, we propose a federated multi-modal segmentation architecture informed by explicit cross-modal feature reasoning. The end-to-end framework comprises five modules: squeeze-excitation module (Fig. 1a), Cross-attention block (Fig. 1b), Multi-modal attention module (Fig. 1c, a hierarchical attention-based fusion mechanism (Fig. 1d). This structure is deployed within a federated learning (FL) protocol that guarantees institutional data privacy. The complete pipeline is shown in Fig. 1e.

For each sample, four complementary MRI modalities—T1, T1CE, T2, and FLAIR—are processed in parallel. The encoder extracts rich modality-specific and shared representations which are carefully fused using paired attention blocks and correlation-aware modules. A decoder with skip connections reconstructs the final segmentation, guided at multiple scales via hierarchical supervision. All learning is performed in a federated setup, aggregating only model weights and ensuring no raw data leaves the local site.

### 2.2 MULTI-MODAL FEATURE EXTRACTION

Let $X = \{X^{T1}, X^{T1CE}, X^{T2}, X^{FLAIR}\}$ denote the set of registered MRI modalities for each case. Each input is a 2D patch $X \in \mathbb{R}^{H \times W \times C}$, where $H$, $W$, $C$ denote the height, width and the number of channels. As we are using a pre-trained network ResNet18 , we repeat the input modalities to have 3 channels, that is, $C = 3$.

Each modality is independently processed through a shared backbone encoder $\mathcal{E}$, which is an ImageNet pre-trained ResNet18 network.

$$F_{en7}^{T1}, F_{en7}^{T1CE}, F_{en7}^{T2}, F_{en7}^{FLAIR} = \mathcal{E}(X^{T1}), \mathcal{E}(X^{T1CE}), \mathcal{E}(X^{T2}), \mathcal{E}(X^{FLAIR}) \tag{2}$$

where $F_{en7}^{T1}, F_{en7}^{T1CE}, F_{en7}^{T2}, F_{en7}^{FLAIR} \in \mathbb{R}^{\frac{H}{32} \times \frac{W}{32} \times 512}$. Additionally, we also take the output from the intermediate layers of the encoder to incorporate into the skip connections of the segmentation architecture. To conform with the standard UNet architecture, we take outputs from the convolutional blocks of the ResNet encoder. To clarify,

$$F_{en2}^{T1}, F_{en2}^{T1CE}, F_{en2}^{T2}, F_{en2}^{FLAIR} \in \mathbb{R}^{\frac{H}{2} \times \frac{W}{2} \times 64} \qquad F_{en4}^{T1}, F_{en4}^{T1CE}, F_{en4}^{T2}, F_{en4}^{FLAIR} \in \mathbb{R}^{\frac{H}{4} \times \frac{W}{4} \times 64}$$

$$\text{(3)} \qquad\qquad\qquad\qquad\qquad\qquad\qquad\qquad\qquad \text{(4)}$$

$$F_{en5}^{T1}, F_{en5}^{T1CE}, F_{en5}^{T2}, F_{en5}^{FLAIR} \in \mathbb{R}^{\frac{H}{8} \times \frac{W}{8} \times 128} \qquad F_{en6}^{T1}, F_{en6}^{T1CE}, F_{en6}^{T2}, F_{en6}^{FLAIR} \in \mathbb{R}^{\frac{H}{16} \times \frac{W}{16} \times 256}$$

$$\text{(5)} \qquad\qquad\qquad\qquad\qquad\qquad\qquad\qquad\qquad \text{(6)}$$

Let $\mathcal{F}_l = \{F_l^{(m)} : m \in \mathcal{M}\}$ denote the set of feature maps at encoding stage $l \in \{en2, en4, en5, en7\}$ for all modalities $m \in \mathcal{M} = \{\text{T1}, \text{T1CE}, \text{T2}, \text{FLAIR}\}$, where $F_l^{(m)} \in \mathbb{R}^{H_l \times W_l \times C_l}$ is the output of the encoder for modality $m$ at level $l$, already defined in Eqn. 3-6.

The outputs from the encoder layers are fed into the attention modules for fusion of the four modalities, as discussed below.

**Positional Embedding:** After the features were extracted from the encoders, we added positional embeddings to the output features from all the modalities. We experimented with both 2D and 3D positional embeddings, and also a combination of both. While 2D positional embeddings give a sense of the spatial relativity of different pixels in the feature map, the 3D positional embeddings were intended to extend this to cross-modal relativistic awareness. The step proceeds as follows,

$$\mathcal{F}_l^m = \mathcal{F}_l^m + PE(\mathcal{F}_l^m) \tag{7}$$

where $PE(\cdot)$ returns the 2D or 3D positional embeddings based on the input shape.

### 2.3 CROSS-MODALITY ATTENTION-BASED FUSION

Recognizing both the shared and unique signatures captured by each MRI sequence, we introduce a Primary-Auxiliary Attention Mechanism (Fig. 2, yellow blocks), which dynamically fuses information among modalities. For each fusion operation, one modality is chosen as "primary" and the rest as "auxiliary." The primary features are augmented using cross-attention modules that exchange critical contextual information from auxiliaries, enhancing the detectability of subtle or noisy boundaries. The cross-attention operation is defined as:

$$\hat{F}^P = \text{CrossAttn}(F^P, \{F^{A_i}\}) \tag{8}$$

where $F^P \in \mathbb{R}^{H \times W \times 3C}$ is the primary feature map and $\{F^{A_i}\}$ are auxiliary streams, where $C$ is the number of channels in each modality feature map. For example, if $F^P$ is the primary feature map from the $T1$ modality, then the feature maps $F^{T1CE}, F^{T2}, F^{FLAIR}$ constitute the auxiliary features maps $\{F^{A_i}\}$. Cross-attention encourages the network to emphasize structures consistently present across modalities and to counteract modality-specific artifacts.

The resulting outputs from the cross-attention modules are concatenated and further processed by a $1 \times 1$ convolutional layer trained to select information from the channels according to their importance, yielding a robust multi-modal embedding $\hat{F}^A$. The output $\hat{F}^A$ is then added to the primary feature map $\hat{F}^P$, which acts as the skip connection, giving the output $\hat{F}'_P$. The final output from the cross-attention module is obtained after passing $\hat{F}'_P$ through a dropout and multi-layered perceptron layer which projects $\hat{F}'_P \in \mathbb{R}^{H \times W \times C_{in}}$ to $F^O \in \mathbb{R}^{H \times W \times C_{out}}$, where $C_{in}$ and $C_{out}$ are the input and output channels of the cross-attention module.

However, this output, $F^O$, is only for one primary feature map, $F^P$. For all four modalities, we set each modality as primary, and get $\{F^{O_i}\}$ where $i \in \{T1, T1CE, T2, FLAIR\}$.

### 2.4 HIERARCHICAL MULTI-MODAL ATTENTION

We propose a hierarchical multi-modal fusion block that explicitly models correlations both within and across MRI modalities at multiple feature scales. This design is motivated by the observations that distinct modalities exhibit complementary sensitivity to tumour subregions and that information at different network depths encodes varying semantic granularity.

#### 2.4.1 MULTI-MODAL CORRELATION-BASED ATTENTION BLOCK

To adaptively weigh and correlate these features, we introduce a Multi-Modal Correlation-based Attention Block $\mathcal{A}_l(\cdot)$ at each hierarchy. For each stage $l$, this block processes the stacked features

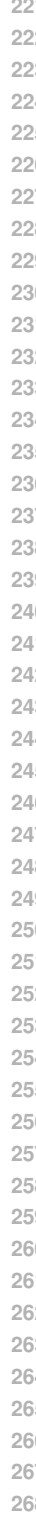

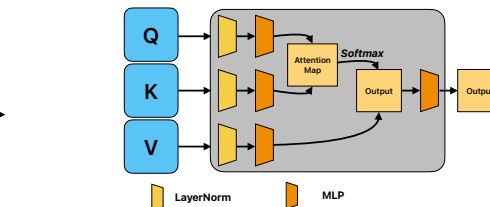

(a) Squeeze-Excitation. $\sigma$: Sigmoid.

(b) Cross-Attention Block

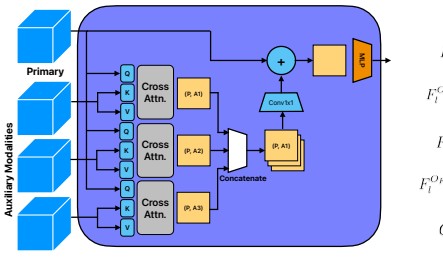

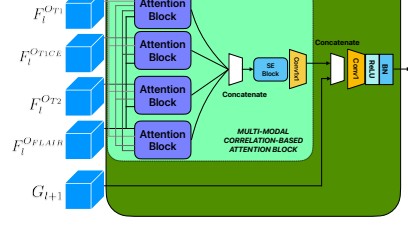

(c) Multi-Modal Attention Module

(d) Hierarchical Fusion and Decoder Integration

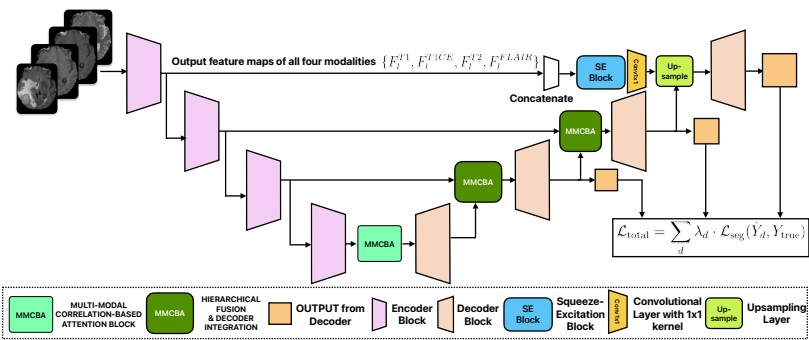

(e) FedUNetMM Pipeline

Figure 1: The above diagrams illustrate the different building blocks of the proposed framework. The diagrams are self-explanatory, and the blocks are colour-coded.

$\{F_l^{O_i}\}$ and yields a modality-fused representation $G_l$. First, correlation attention weights $W_l \in [0, 1]^{4C_l}$ are computed as:

$$S_l = \text{AvgPool}(\hat{F}_l^{\text{joint}}) \in \mathbb{R}^{1 \times 1 \times 4C_l} \tag{9}$$

where $\hat{F}_l^{\text{joint}} = \text{Concatenate}(F_l^{O_{T1}}, F_l^{O_{T1CE}}, F_l^{O_{T2}}, F_l^{O_{FLAIR}})$. The output $S_l$ is further processed through two convolutional layers with a ReLU activation layer in between, as follows,

$$S_l' \leftarrow \sigma(\text{Conv2}(\text{LeakyReLU}(\text{Conv1}(S_l)))) \tag{10}$$

where $\text{Conv1}, \text{Conv2}$ are $1 \times 1$ convolutions and $\sigma$ denotes the element-wise sigmoid. The recalibrated feature is produced by

$$\tilde{F}_l = \hat{F}_l^{\text{joint}} \odot W_l \tag{11}$$

where $\odot$ denotes channel-wise multiplication. The final output $G_l$ is obtained by passing $\hat{F}_l$ through a final $1 \times 1$ convolutional layer.

We apply the cross-modality attention-based fusion block to the output feature maps from every layer except at the layer $l = en2$, as it causes the memory to be exceeded, resulting in failure of training. For $l = en2$, we simply concatenate the features $F_{en2}^{T1}, F_{en2}^{T1CE}, F_{en2}^{T2}, F_{en2}^{FLAIR}$ to obtain $\hat{F}_{en2}^{joint}$. The rest of the process remains the same to obtain $G_{en2}$.

### 2.4.2 Hierarchical Fusion and Decoder Integration

The features $\{G_l\}$ from different encoder stages are propagated to the decoder via skip connections. To facilitate multi-scale integration, we concatenate the fused output at each decode stage $d$ with corresponding encoder features:

$$H_d = \text{Concat}(G_d, D_{d+1}) \tag{12}$$

where $D_{d+1}$ is the upsampled decoder feature from the previous scale.

We then perform a $1 \times 1$ convolution:

$$D_d = \text{BN}(\text{ReLU}(\text{Conv1x1}(H_d))) \tag{13}$$

This decoder output is then passed on to the next layer for further decoding till the final stage.

### 2.4.3 Deep Supervision

For stabilization and improved gradient flow, we introduce auxiliary segmentation heads at each decode scale:

$$\hat{Y}_d = f_{\text{seg},d}(D_d) \tag{14}$$

where $f_{\text{seg},d}$ is a segmentation head. The segmentation heads were designed as,

$$f_{seg,d} = \text{Conv3}(\text{BN}(\text{ReLU}(\text{Conv2}(\text{BN}(\text{ReLU}(\text{Conv1}(\cdot)))))) \tag{15}$$

where Conv1, Conv2 and Conv3 are all convolutional layers with kernel $3 \times 3$.

### 2.5 Decoder and Deep Supervision

The unified decoder reconstructs segmentation masks from the fused multi-scale, multi-modal representations, employing standard upsampling and convolution operations augmented with skip connections from the encoder. To mitigate potential information loss and reinforce gradient propagation during training, deep supervision is applied: intermediate predictions are generated at multiple decoder stages, each supervised with the ground truth maps.

The final output is composed via a concatenation of multi-scale, multi-modal predictions, followed by a final classification layer. However, during inference, only the final classification layer output is used.

### 2.6 Implementation Details and Loss

All models are optimized using a hybrid loss combining Dice and binary cross-entropy, standardizing scale across deep supervision heads. The architecture is implemented in PyTorch. All the client models were trained for 30 communication rounds, each with 3 local epochs. We used the Adam optimizer for optimizing the parameters in all the clients, with an initial learning rate of $1e - 4$, and a cosine learning rate decay scheduler. We limited the batch size to 8 due to a limited computational constraints scenario of our federated learning framework. We also augment the input MR slices by randomly cropping with dimensions $224 \times 224$, random rotation and flipping, and random noise addition with standard deviation $0.01$. The model parameters were updated by optimizing the following loss:

$$\mathcal{L}_{\text{total}} = \sum_d \lambda_d \cdot \mathcal{L}_{\text{seg}}(\hat{Y}_d, Y_{\text{true}}) = \sum_d \lambda_d \cdot \left( \mathcal{L}_{CE}(\hat{Y}_d, Y_{\text{true}}) + \mathcal{L}_{Dice}(\hat{Y}_d, Y_{\text{true}}) \right) \tag{16}$$

where $\lambda_d$ are scale weights, and calculated based on the ratio of the output dimension to the final output dimension, that is $\lambda_d = \frac{H_d}{224}$, where $H_d$ is the height of the $d$-th decoder layer output.

The comparative baselines were trained for 500 epochs from scratch with a batch size of 2, using the Adam optimizer and an initial learning rate of $1e - 4$ with a cosine learning rate decay schedule.

In summary, our methodology couples a federated approach with modality- and hierarchy-aware attention mechanisms, aligning with recent advances in the field while addressing the unique challenges of secure, multi-modal clinical imaging. This structured fusion enables the model to effectively harness the full spectrum of multi-institutional, multi-modal data, yielding robust and generalizable brain tumour segmentation.

# 3 EXPERIMENTAL DETAILS AND RESULTS

## 3.1 DATASET

To evaluate the efficacy and generalizability of our federated multi-modal segmentation framework, we utilize the multi-institutional Brain Tumour Segmentation (BraTS) benchmark datasets from the years 2018, 2019, and 2020. Each case comprises four co-registered modalities: T1, T1-weighted contrast-enhanced (T1CE), T2, and Fluid-Attenuated Inversion Recovery (FLAIR).

BraTS2018 contains 285 training cases, BraTS2019 includes 335 cases, and BraTS2020 consists of 369 cases, all provided in the NIfTI format. Annotations for Enhancing Tumour (ET), Tumour Core (TC), and Whole Tumour (WT)—were delineated by expert neuro-radiologists. The WT consists of Necrosis Core (NEC), ET and Edema (ED), and TC consists of NEC and ET.

**Data Distribution:** In our federated training protocol, the aggregated data pool from all three BraTS cohorts is partitioned among $N = 20$ clients, simulating a multi-institutional distributed learning scenario. To model non-identical and unbalanced data distributions, we employ a Dirichlet allocation scheme with a concentration parameter $\alpha = 1.0$, following standard federated learning practices. Under this scheme, each client $k$ receives a unique subset $D(k)$ consisting of both the number and class proportion of segmentation samples drawn independently from a Dirichlet($\alpha$) distribution over tumour categories. This results in heterogeneous and possibly imbalanced data assignments, more closely mirroring clinical environments where institutional data distributions may diverge due to cohort, scanner protocol, or pathology prevalence.

Mathematically, let there be $M$ total patients in the combined dataset and $N$ federated clients. To simulate non-IID data distributions across institutions, for each client $k \in \{1, \ldots, N\}$, we first sample a proportion vector $(q_1, q_2, \ldots, q_N) \sim$ Dirichlet($\alpha \cdot 1_N$), with $\alpha$ set to 1.0. The patient list is then randomly shuffled, and approximately $q_k \cdot M$ patients are assigned to client $k$. Thus, the assignment of patients to clients follows a Dirichlet($\alpha$) distribution at the patient (case) level, ensuring heterogeneous and imbalanced partitions across sites.

Under this assignment, some clients may receive more or fewer patients, and the local composition of tumour types and modalities reflects the natural variability of the data, mirroring real-world settings. The Dirichlet allocation satisfies the constraint $\sum_{k=1}^{N} q_k = 1$.

**Centralized Test Set:** To enable robust and consistent model evaluation, a fixed portion of the dataset is withheld from the federated clients and reserved as a centralized test set on the server. For our experiments, we randomly sampled 10% samples from BraTS2018, BraTS2019, and BraTS2020, respectively, and used those samples as the centralized test set.

Table 1: Comparison on BraTS 2018 dataset based on Dice scores (%). Best results in bold.

| Methods | Dice (%) | | | | Params (M) | FLOPs (G) |
|---|---|---|---|---|---|---|
| | ET | TC | WT | Average | | |
| 3D U-Net [6] | 73.88 | 81.32 | 89.71 | 81.64 | 16.32 | 1899.47 |
| Residual 3D U-Net [35] | 72.80 | 79.87 | 89.89 | 80.83 | 113.74 | 2601.05 |
| TransBTS w/o TTA [14] | 69.36 | 77.47 | 87.09 | 77.97 | 14.53 | 219.92 |
| TransBTS w/ TTA [14] | 71.05 | 81.20 | 88.07 | 80.11 | 32.99 | 263.73 |
| UNETR [15] | 71.01 | 78.99 | 89.77 | 79.92 | 97.73 | 292.18 |
| Swin UNETR [26] | 70.71 | 74.24 | **89.01** | 77.32 | 62.65 | 194.02 |
| UNETR++ [16] | 71.05 | 81.20 | 88.07 | 80.11 | 42.65 | 139.02 |
| nnFormer [17] | 70.17 | 79.76 | 89.96 | 79.96 | 49.66 | 372.06 |
| UNetMM (Centralized) | 77.76 | 84.56 | 84.75 | 82.37 | 20.5 | 69.01 |
| FedUNetMM | **76.93** | **84.35** | 84.63 | **82.25** | 20.5 | **69.01** |

Table 2: Comparison on BraTS 2019 dataset based on Dice scores (%). Best results in bold.

| Methods | Dice (%) | | | | Params (M) | FLOPs (G) |
|---|---|---|---|---|---|---|
| | ET | TC | WT | Average | | |
| 3D U-Net [6] | 73.58 | 81.31 | 89.61 | 81.48 | 16.32 | 1899.47 |
| Residual 3D U-Net [35] | 69.41 | 76.55 | 89.46 | 78.47 | 113.74 | 2601.05 |
| TransBTS w/o TTA [14] | 70.53 | 76.32 | 88.81 | 78.55 | 14.53 | 219.92 |
| TransBTS w/ TTA [14] | 70.73 | 76.99 | **90.13** | 79.28 | 32.99 | 263.73 |
| UNETR [15] | 70.57 | 74.22 | 89.92 | 78.24 | 97.73 | 292.18 |
| Swin UNETR [26] | 69.99 | 71.94 | 88.82 | 76.25 | 62.65 | 194.02 |
| UNETR++ [16] | 68.57 | 76.08 | **90.13** | 78.26 | 42.65 | 139.02 |
| nnFormer [17] | 67.59 | 75.04 | 87.06 | 76.56 | 49.66 | 372.06 |
| UNetMM (Centralized) | 77.23 | 84.83 | 86.74 | 82.93 | 20.5 | 69.01 |
| FedUNetMM | **76.88** | **84.63** | 86.38 | **82.63** | 20.5 | **69.01** |

Table 3: Comparison on BraTS 2020 dataset based on Dice scores (%). Best results in bold.

| Methods | Dice (%) | | | | Params (M) | FLOPs (G) |
|---|---|---|---|---|---|---|
| | ET | TC | WT | Average | | |
| 3D U-Net [6] | 72.49 | 78.47 | 90.74 | 80.57 | **16.32** | 1899.47 |
| Residual 3D U-Net [35] | 73.72 | 78.43 | 90.10 | 80.75 | 113.74 | 2601.05 |
| TransBTS w/o TTA [14] | 71.07 | 77.02 | 89.59 | 79.23 | 32.99 | 263.73 |
| TransBTS w TTA [14] | 73.09 | 79.20 | 91.12 | 81.14 | 32.99 | 263.73 |
| UNETR [15] | 69.97 | 76.60 | 88.50 | 78.36 | 102.12 | 179.78 |
| SwinUNETR [26] | 72.18 | 79.59 | **91.87** | 81.21 | 62.19 | 794.02 |
| UNETR++ [16] | 74.30 | 77.49 | 91.03 | 80.94 | 42.65 | **139.41** |
| nnFormer [17] | 71.45 | 79.57 | 91.80 | 80.94 | 149.68 | 372.06 |
| UNetMM (Centralized) | 76.56 | 83.02 | 88.53 | 82.70 | 20.5 | 69.01 |
| FedUNetMM | **75.87** | **83.17** | 88.44 | **82.49** | 20.5 | **69.01** |

## 3.2 QUANTITATIVE RESULTS

The experimental segmentation performance for the proposed framework on the three datasets, BraTS2018, BraTS2019, and BraTS2020, is presented in Table 1, 2, and 3, respectively.

**Interpretation:** Despite lacking explicit volumetric context, our 2D multimodal attention design has preferentially benefitted the ET and TC subregions. ET delineation is strongly driven by T1CE contrast; the cross-modal attention amplifies and enhances rims and suppresses modality-specific noise, yielding sharper ET masks. TC, defined as ET plus necrotic core, likewise profits from high in-plane resolution and cross-modality correlation, which sharpen boundaries and capture compositional cues. By contrast, WT depends more on diffuse edema and continuity across slices, where 3D context is typically advantageous; this helps explain our comparatively smaller gains on WT.

From a federated perspective, a 2D backbone markedly reduces client-side memory/compute, enabling larger effective batch sizes and faster local epochs; together with ImageNet pretraining, this lowers the number of communication rounds needed to reach a target performance. Our results should therefore be interpreted as evidence that, under realistic constraints, a well-designed, pretrained 2D multimodal model is a strong and efficient alternative—particularly for ET/TC—rather than a blanket replacement for volumetric approaches.

### 3.2.1 ABLATION ON POSITION EMBEDDINGS (PE)

We investigate how spatial priors affect our model via an ablation on position embeddings. We compare our model both without and with 2D sinusoidal embeddings and 3D sinusoidal embeddings in Table 4. For 3D PE, we concatenate the feature maps of each modality from the encoder and treat them as a 3D volume. After adding the positional embeddings, the feature maps for each modality are again separated. We also use 2D and 3D PE together. However, we do not observe any significant variation in the prediction scores, as established through a lower one-tailed chi-squared test with $s_0 = 2.0$, where $s_0$ is the standard deviation threshold, for which we obtained $p < 0.05$ for all three classes. The normality of the prediction scores is evident through the Kolmogorov-Smirnov Test.

## 3.3 QUALITATIVE RESULTS

We present representative qualitative results on the centralized test set spanning BraTS 2018–2020, selecting both typical and challenging cases from the non-IID federated splits, in Fig. 2. We plot

Table 4: Comparison of the combination of 2D and 3D positional encoding in the BraTS 2018 datasets. 'Last' and 'All' denote the positional embedding being applied to feature maps from the last layer and all layers of the encoder.

| Methods | Positional Encoding | | | | Dice (%) | | | |
|---|---|---|---|---|---|---|---|---|
| | 2D | | 3D | | ET | TC | WT | Average |
| | Last | All | Last | All | | | | |
| FedUNetMM | ✓ | × | × | × | **77.76** | 84.35 | 84.63 | **82.25** |
| FedUNetMM | ✓ | ✓ | × | × | 76.89 | 84.17 | **84.76** | 81.94 |
| FedUNetMM | × | × | ✓ | × | 76.77 | **84.42** | 83.79 | 81.66 |
| FedUNetMM | × | × | ✓ | ✓ | 76.63 | 84.26 | 83.97 | 81.62 |
| FedUNetMM | ✓ | ✓ | ✓ | ✓ | 76.51 | 84.31 | 84.23 | 81.68 |

each of the classes Necrotic Core (NEC), Edema (ED), and Enhancing Tumor (ET) separately. The odd rows represent the ground truths, whereas the even rows exhibit the predicted masks.

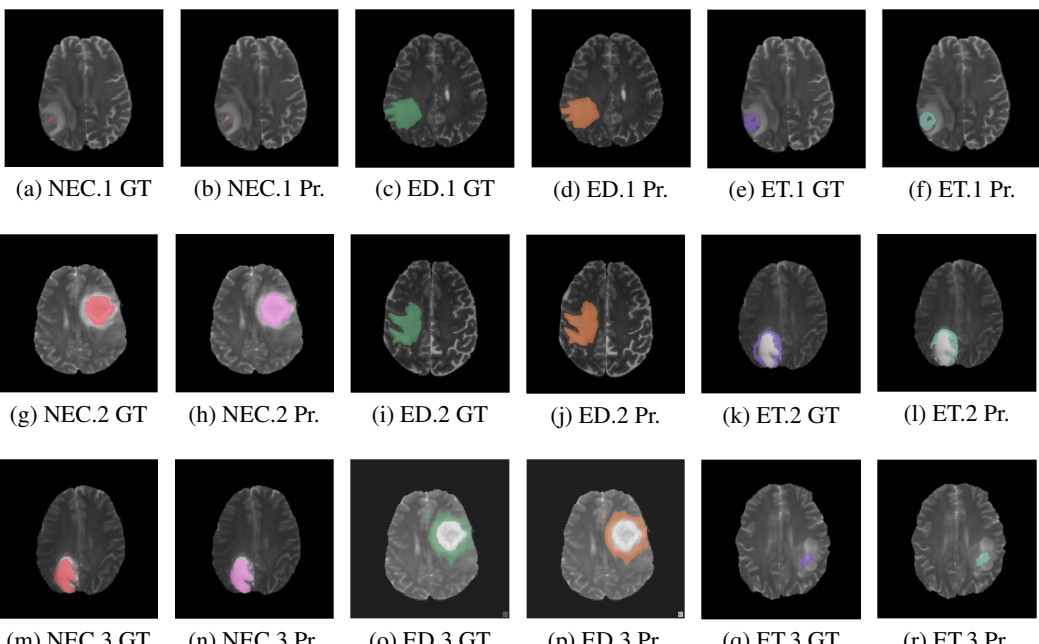

(a) NEC.1 GT    (b) NEC.1 Pr.    (c) ED.1 GT    (d) ED.1 Pr.    (e) ET.1 GT    (f) ET.1 Pr.

(g) NEC.2 GT    (h) NEC.2 Pr.    (i) ED.2 GT    (j) ED.2 Pr.    (k) ET.2 GT    (l) ET.2 Pr.

(m) NEC.3 GT    (n) NEC.3 Pr.    (o) ED.3 GT    (p) ED.3 Pr.    (q) ET.3 GT    (r) ET.3 Pr.

Figure 2: Visual samples of the three classes NEC, ED and ET. The odd and even columns denote the ground truth (GT) and predicted (Pr.) masks, respectively. Different colours for better visualization.

## 4 CONCLUSION

In this work, we proposed a federated learning framework for multi-modal brain tumor segmentation, integrating a hierarchical multi-modal correlation attention module and leveraging the FedAvg algorithm for collaborative model training across simulated medical institutions. By utilizing BraTS 2018, 2019, and 2020 datasets and partitioning the cases from each of those datasets among 20 clients according to a Dirichlet distribution, we closely mirrored the heterogeneity and statistical challenges of real-world clinical data sharing scenarios. Our experiments demonstrate that the hierarchical attention mechanism effectively fuses rich feature representations across MRI modalities and spatial scales, yielding robust and generalizable segmentation performance. Extensive evaluation on centralized test set showcases strong generalization to unseen cases. Overall, our study underscores the feasibility and advantages of federated collaborative learning for large-scale, privacy-sensitive neuroimaging applications by enabling faster convergence and lower computational overhead. Future work will explore more advanced aggregation algorithms or a feature extraction pipeline to further strengthen segmentation performance in federated medical AI.

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
