# A   FEDAVG PSEUDO-CODE

---

**Input:** Number of communication rounds $T$; number of clients $K$; local epochs $E$; learning rate $\eta$; Initial global model weights $w^0$

---

**for** $t = 0, 1, \ldots, T-1$ **do**
    Server broadcasts $w^t$ to all participating clients;
    **for** *each client $k \in \{1, \ldots, K\}$* **do in parallel**
        $w_k^{t,0} \leftarrow w^t$;
        **for** $e = 1, \ldots, E$ **do**
            **for** *each minibatch b in local data $\mathcal{D}^{(k)}$* **do**
                $wk^{t,e} \leftarrow wk^{t,e-1} - \eta \nabla \mathcal{L}^{(k)}(w_k^{t,e-1}; b)$;
            **end**
        **end**
        Client $k$ sends updated weights $w_k^{t,E}$ to server;
        Let $n_k$ be the local sample size for client $k$;
    **end**
    Server aggregates updates:
        $ntot = \sum k = 1^K n_k$;
        $w^{t+1} = \sum k = 1^K \frac{nk}{ntot} wk^{t,E}$;
**end**
**Output:** Final global model weights $w^T$

---

**Algorithm 1:** Federated Training Protocol with FedAvg