# OpenReview forum: "Federated Brain Tumour Segmentation using Multi-Modal Information Fusion"
_ICLR.cc/2026/Conference — ICLR 2026 Conference Withdrawn Submission_

### Official Review · Reviewer_p3qh · 2025-10-29

**Soundness:** 3
**Presentation:** 3
**Contribution:** 2
**Rating:** 2
**Confidence:** 4

**Summary:**

This paper proposes a federated learning framework for multi-modal brain tumor segmentation that integrates hierarchical cross-modal fusion. The method is designed to mitigate non-iid client distributions and communication constraints commonly encountered in federated medical imaging. Evaluation on the BraTS dataset shows improvements over several compared methods and performance close to a centralized upper bound.

**Strengths:**

-	Brain tumor segmentation is clinically significant, and the application of federated learning aligns with privacy and data governance needs in healthcare.
-	The paper is well organized; the method and experimental setup are easy to follow.
-	The proposed approach achieves results comparable to centralized training and outperforms multiple baseline federated methods.

**Weaknesses:**

-	This work presents a solid application of federated learning for multi-modal brain tumor segmentation, with technical details and experimental results clearly reported. However, from a technical and theoretical standpoint, the paper may not meet the publication standards expected by this community. The design of the model architecture—such as the use of multi-modal attention and hierarchical fusion—is relatively straightforward and appears to emphasize engineering implementation over novel research contributions. The specific technical innovations and their significance should be further clarified and better articulated to establish a stronger research contribution.
-	The paper would further benefit from theoretical analysis or guarantees (e.g., convergence under modality-skewed non-iid, effects of sparsified updates)
-	The current evaluation focuses on BraTS. A more realistic federated setup (asynchrony, client dropouts, heterogeneous hardware, missing modalities, privacy accounting) and discussion of clinical translation would greatly strengthen the paper.
-	It would be helpful to detail how non-iid is constructed (site-, pathology-, protocol-based splits), report divergence metrics, and analyze performance sensitivity to non-iid severity.

**Questions:**

I have no specific questions. To further enhance this work, the authors could consider the following studies: a. communication efficiency: Provide per-round and total communication costs, compression ratios, and convergence rounds. Compare accuracy-cost trade-offs against strong baselines with similar budgets. b. Robustness to missing/partial modalities: Include experiments where one or more sequences are unavailable or corrupted, and show how the model handles dynamic modality masking at inference. c. Generalization: Further evluating the model for unseen cohorts. d. Privacy and security: Consider using secure aggregation and/or differential privacy to make this framework comprehensive.

---

### Official Review · Reviewer_Qr47 · 2025-10-31

**Soundness:** 3
**Presentation:** 2
**Contribution:** 2
**Rating:** 2
**Confidence:** 3

**Summary:**

This research introduces FedUNetMM, a federated learning system for multi-modal brain tumor segmentation with MRI modalities (T1, T1CE, T2, and FLAIR).  The approach uses a hierarchical attention-based multi-modal fusion mechanism and a cross-modality correlation-aware attention module to successfully combine modality-specific and shared representations.

**Strengths:**

1. Cross-modal attention, hierarchical fusion, and deep supervision are all integrated into a single FL framework.

2. Consistently competitive with centralized baselines on BraTS 2018-2020, notably in ET and TC segmentation.

3. Mathematical formulations and architecture schematics are thorough, repeatable, and follow recognized FL principles.

**Weaknesses:**

1, There is little innovation in the fundamental FL approach; FedAvg is used without any improvements or comparison to adaptive or privacy-preserving versions.

2. There was no ablation on FL components, including client count, non-IID severity (α variation), and communication cost.

3. Minor overemphasis on architectural elements, with little explanation of how FL dynamics interact with multi-modal fusion.

**Questions:**

1. What is the novelty in this paper that closely follows the approaches of ICLR papers?

2. Could the model manage missing modalities at some clients, which is a typical problem in actual multi-institutional settings?

---

### Official Review · Reviewer_Ngzg · 2025-11-01

**Soundness:** 2
**Presentation:** 2
**Contribution:** 2
**Rating:** 2
**Confidence:** 4

**Summary:**

This paper proposes FedUNetMM, a federated learning framework for multi-modal brain tumor segmentation that combines 2D U-Net with cross-modal attention mechanisms. It uses four MRI modalities (T1, T1CE, T2, FLAIR) and integrates multi-modal fusion and deep supervision.

**Strengths:**

1、Applies federated learning to a clinically relevant and privacy-sensitive application, which aligns with the growing demand for privacy-preserving AI in healthcare.
2、Proposes a multi-modal fusion strategy that enhances segmentation performance on challenging tumor subregions (ET and TC), demonstrating the potential benefit of cross-modal attention.
3、The experimental setup and training details are described with reasonable clarity.
4、The paper concludes with a coherent summary that connects the empirical findings to the overall objectives.

**Weaknesses:**

1、Motivation and novelty are unclear. The Introduction emphasizes general challenges in federated learning, but the paper does not sufficiently target or experimentally address these specific issues. Using FedAvg as the aggregation strategy is standard; the claimed use of pretrained initialization to speed convergence is practical but not a fundamental methodological contribution.
2、Limited contribution. The proposed 2D approach inherently lacks volumetric context compared to 3D networks (e.g., 3D U-Net), which is important for whole-tumor delineation and boundary accuracy. The manuscript acknowledges this limitation but fails to investigate it deeply or to provide a comprehensive comparison against established 3D baselines. It is therefore unclear whether the fusion strategy leads to clinically meaningful improvements.
3、Insufficient comparative and ablation studies. Quantitative comparisons are provided, but the ablation studies and detailed analyses are shallow. The federated experiments span multiple BraTS versions, yet the paper lacks comparisons with alternative federated algorithms that are designed to handle non-IID data.
4、Communication cost and deployment analysis are inadequate. The authors claim reduced communication rounds and faster convergence with pretraining, but they do not present a detailed accounting of communication overhead.
5、Limited evaluation metrics. The evaluation relies only on region-based Dice scores. Boundary and surface metrics such as HD95 and ASSD are missing but important for segmentation quality assessment in clinical contexts.
6、Missing figure reference. Figure 2 is referenced around line 185 but cannot be found.

**Questions:**

1、What is the primary, concrete motivation for the proposed work—i.e., which specific practical problem in federated medical imaging does this method solve better than prior art?
2、The paper promotes a multi-modal fusion strategy. How does the method behave under modality dropout (missing modalities)? Compared to single-modality baselines, what are the quantitative gains when one or more modalities are missing, and are these gains clinically significant?
3、Have you experimented with other federated learning paradigms ? If so, how do they compare to FedAvg in your setup?

---

### Official Review · Reviewer_bYv8 · 2025-11-04

**Soundness:** 2
**Presentation:** 2
**Contribution:** 2
**Rating:** 2
**Confidence:** 4

**Summary:**

This paper presents FedUNetMM, a federated multimodal framework for brain tumor segmentation. The model is based on a lightweight 2D ResNet-18 U-Net and incorporates cross-modal attention mechanisms, hierarchical correlation attention modules, and deep supervision. Experiments were conducted on the BraTS 2018/2019/2020 datasets using 20 non-independent and identically distributed clients. The results show that the federated model achieves nearly the same Dice score as the centralized model while reducing computational and communication costs.

**Strengths:**

- The model has a clear structure and is well-designed; the attention module can effectively integrate information from different magnetic resonance -imaging (MRI) modalities.
- The experiments are comprehensive, covering multiple datasets and non-IID settings.
- The model is proposed to be lightweight and suitable for practical federated learning scenarios.

**Weaknesses:**

- Lack of Optimization in Federated Learning: The authors mostly focus on designing their UNetMM architecture and do not propose any optimizations for federated learning. They only use the FedAvg method, so the small computation and communication costs shown in the experiments are due to the small task model, rather than any federated learning design. Therefore, I believe this work has little to do with federated learning. The authors are primarily designing a lightweight Hierarchical Cross-Modal Attention Fusion technique and only attempted federated learning under FedAvg. As a result, the claimed contributions to federated learning design are not clearly demonstrated.
- Lack of novelty: The authors do not provide a detailed explanation of each design decision for their architecture, and the design points are not validated experimentally. The architecture is based on traditional U-Net, and the attention design is very similar to UNetR++, with the main difference being the use of 2D attention. The so-called multi-modal fusion involves parallel inputs of multiple modalities into the model. Therefore, the entire algorithm process seems like a minor innovation in U-Net, with limited novelty.
- Experimental Results Do Not Strongly Validate the Method's Advantages: The experimental results do not provide strong validation for the advantages of the proposed method. From Tables 2 and 3, we can see that the metrics for WT are much lower than all the baselines. In Table 4, different types of PE yield varying effects, and using all of them together leads to worse performance. Which one is optimal, and why? This lack of clarity in the results weakens the overall argument.

**Questions:**

- What is the rationale behind each part of the architecture design? I would like the authors to provide a detailed explanation of the design decisions for each part of the architecture. Additionally, it would be helpful if they could validate these design choices through experimental results.
- Are there any optimization designs related to federated learning? Does the proposed method include any specific optimizations for federated learning, aside from simply using FedAvg? It would be beneficial if the authors could explain any novel federated learning techniques they have employed.
- Can the experiments present more federated learning-related variables? In the experiments, can the authors include more federated learning-specific metrics, such as communication costs, computation time, or other relevant factors?

---

### Note · Authors · 2025-12-10

I have read and agree with the venue's withdrawal policy on behalf of myself and my co-authors.